# Synthesis, Characterization and Potential Antimicrobial Activity of Selenium Nanoparticles Stabilized with Cetyltrimethylammonium Chloride

**DOI:** 10.3390/nano13243128

**Published:** 2023-12-13

**Authors:** Anastasiya Blinova, Andrey Blinov, Alexander Kravtsov, Andrey Nagdalian, Zafar Rekhman, Alexey Gvozdenko, Maksim Kolodkin, Dionis Filippov, Alina Askerova, Alexey Golik, Alexander Serov, Mohammad Ali Shariati, Naiyf S. Alharbi, Shine Kadaikunnan, Muthu Thiruvengadam

**Affiliations:** 1Physical and Technical Faculty, North-Caucasus Federal University, 355017 Stavropol, Russia; aablinova@ncfu.ru (A.B.); avblinov@ncfu.ru (A.B.); akravtcov@ncfu.ru (A.K.); zaarekhman@ncfu.ru (Z.R.); agvozdenko@ncfu.ru (A.G.); mkolodkin@ncfu.ru (M.K.); ddfilippov@ncfu.ru (D.F.); abgolik@ncfu.ru (A.G.); 2Laboratory of Food and Industrial Biotechnology, North-Caucasus Federal University, 355017 Stavropol, Russia; vikalinka04@mail.ru; 3Chemical and Pharmaceutical Faculty, North-Caucasus Federal University, 355017 Stavropol, Russia; aserov@ncfu.ru; 4Scientific Department, Semey Branch of the Kazakh Research Institute of Processing and Food Industry, Gagarin Avenue 238G, Almaty 050060, Kazakhstan; shariatymohammadali@gmail.com; 5Department of Botany and Microbiology, College of Science, King Saud University, P. O. Box 2455, Riyadh 11451, Saudi Arabia; nalharbi1@ksu.edu.sa (N.S.A.); sshine@ksu.edu.sa (S.K.); 6Department of Applied Bioscience, Konkuk University, Seoul 05029, Republic of Korea

**Keywords:** selenium nanoparticles, CTAC, ζ-potential, transmission electron microscopy, stability, antibacterial properties, fungicidal properties

## Abstract

Selenium nanoparticles (Se NPs) have a number of unique properties that determine the use of the resulting nanomaterials in various fields. The focus of this paper is the stabilization of Se NPs with cetyltrimethylammonium chloride (CTAC). Se NPs were obtained by chemical reduction in an aqueous medium. The influence of the concentration of precursors and synthesis conditions on the size of Se NPs and the process of micelle formation was established. Transmission electron microscopy was used to study the morphology of Se NPs. The influence of the pH of the medium and the concentration of ions in the sol on the stability of Se micelles was studied. According to the results of this study, the concentration of positively charged ions has a greater effect on the particle size in the positive Se NPs sol than in the negative Se NPs sol. The potential antibacterial and fungicidal properties of the samples were studied on *Escherichia coli*, *Micrococcus luteus* and *Mucor*. Concentrations of Se NPs stabilized with CTAC with potential bactericidal and fungicidal effects were discovered. Considering the revealed potential antimicrobial activity, the synthesized Se NPs-CTAC molecular complex can be further studied and applied in the development of veterinary drugs, pharmaceuticals, and cosmetics.

## 1. Introduction

A crucial component for both human and animal health is selenium (Se). It is a component of Se-cysteine, an amino acid needed to create a number of selenium-proteins in the human body [1]. Se deficiency in both humans and animals leads to serious problems such as hypothyroidism, Keshan and Kashin–Beck diseases [2,3,4,5]. Eliminating Se deficit is possible and realistic with biofortification. Profiting from nanotechnology, selenium nanoparticles (Se NPs) have recently been proved to show stronger biological activity and lower toxicity than the traditional selenium compounds [6,7,8]. Further, Se NPs are able to inhibit the growth of microorganisms [9,10].

The particle form, size, and surface chemistry have been recognized as crucial parameters determining the interaction of nanomaterials with biological entities [11]. However, Se NPs, despite their advantages and unique properties, tend to aggregate into dark clusters due to their high surface energy [12]. Such aggregates are thermodynamically stable but biologically inert [13]. Consequently, disperse state is one of the critical parameters that determine biologically activity of Se NPs and their industrial application [14]. In this regard, it is necessary to take into account possible ways of obtaining Se NPs, as well as to study the issue of their stability, physicochemical and dispersion properties.

In practice, both biological and chemical reductions can be used to conditionally classify Se NPs synthesis techniques [15,16]. Through the use of biological agents such as bacteria or plant extracts, various inorganic and chemical compounds of Se can be reduced to convenient and non-toxic Se NPs. For example, extracts of *Allium cepa*, *Malpighia emarginata* and *Gymnanthemum amygdalinum* were used to generate Se NPs [17,18,19]. The obtained nanoparticles were measured between 245 and 321 nm. In addition, other researchers generated Se NPs using starch encapsulation [20], bacteria (saltophilic bacteria) [21] and orange peel waste [22]. Chemical reduction uses compounds that have a reducing property to Se. Particle size is controlled using surfactants or growth-preventing reagents such as polyvinyl chloride (PVC) and folic acid. Polyvinylpyrrolidone [23], polyvinyl alcohol [24], protein molecules [25] and polysaccharides [26] can be used as stabilizers for Se NPs. Among them, polysaccharides are considered as an appropriate option for fabricating Se NPs when energy efficiency and eco-friendliness are needed [26,27]. However, Se NPs stabilized with polysaccharides have still unacceptable instability, due to enlargement of size and decline of zeta potential during preservation in aqueous environment [27,28].

Nowadays, surfactants have been widely used to coat nanoparticles for biomedical use [29,30]. Quaternary ammonium compounds are typical surfactants with a hydrophilic ammonium head group and one or two hydrophobic tail groups. Due to their amphoteric nature, they have antiseptic, anesthetic and antibacterial properties [31]. Quaternary ammonium compounds are used in the manufacture of various cosmetic products [32], to form micelles in drug delivery systems (especially in tumor therapy) and as a stabilizer for nanoparticle synthesis [33]. The environmental safety and low toxicity of these substances determine their use in various areas [34]. In contrast, Se NPs can be used in the veterinary, pharmaceutical and cosmetic industries due to their low toxicity, higher absorption capacity, and frequent elimination from the body [34,35,36,37]. Currently, Se NPs are widely applied in medical diagnosis and drug delivery [38,39,40]. The synthesis and study of Se NPs is a topical issue regarding their useful functions and properties [35,36]. Considering the high activity and low stability of Se NPs, the use of ionic and non-ionic surfactants as stabilizers can provide their practical application [37,38].

Nanomaterials have enzymatic catalytic properties and accelerate biocatalytic reactions. Unlike natural enzymes, nanosystems are highly stable and quite easy to obtain. Thus, nanomaterials are widely used in fields such as bacterial theranostics and as antibacterial agents [41,42]. Filipović et al. [11] reported that the use of Se NPs as antimicrobial agents is a promising strategy, especially when dealing with chronic and nosocomial infections. Generally, various mechanisms of nanoparticles antimicrobial activity were recognized to date: ROS generation, interaction with cell barrier (cell wall disruption and alteration in permeability), inhibition in the synthesis of proteins and DNA, expression of metabolic genes, etc. [43,44,45]. The main advantage of Se NPs as antimicrobial agents could be their ability to simultaneously act through these multiple mechanisms, which can solve the problem of the development of multidrug-resistant bacterial strains tolerant to commercial antibiotics [46]. 

Considering the above issues, the aim of this work was to synthesize and characterize Se NPs stabilized with cetyltrimethylammonium chloride (CTAC) and to study their potential antimicrobial activity towards *Escherichia coli*, *Micrococcus luteus* and *Mucor*.

## 2. Materials and Methods

### 2.1. Synthesis of Se NPs-CTAC

Se NPs were synthesized by chemical reduction in aqueous media in the presence of a stabilizer [47]. CTAC (analytical grade, VitaReaktiv, Dzerzhinsk, Russia) was used as the stabilizer. Selenious acid (analytical grade, INTERKHIM, Moscow, Russia) was used as a Se-containing precursor. Ascorbic acid (analytical grade, LenReaktiv, Saint Petersburg, Russia) was used as the reducing agent.

The synthesis of Se NPs was carried out as follows. Solutions were prepared with different ratios (from 0.5 to 4) of the amount of quaternary ammonium compound and the amount of selenious acid. For this purpose, from 0.68 g to 5.24 g CTAC was dissolved in 100 mL of 0.036 M selenious acid solution, depending on the ratio specified. 0.088 M ascorbic acid solution was prepared by dissolving 773.8 mg of acid in 50 mL of distilled water. At the third stage, a solution of ascorbic acid was added dropwise to solutions of selenious acid and CTAC, stirring vigorously. The obtained sample was stirred for 5–10 min. The schematic diagram of the synthesis of Se NPs-CTAC is shown in Figure 1.

### 2.2. Optimization of the Synthesis of Se NPs-CTAC

To optimize the experimental settings, a multivariate experiment with three input parameters and three levels of variation was carried out. The average hydrodynamic particle radius (r_av_) and ζ-potential energy were the output parameters. The degree of fluctuation of the parameters was assessed using early experiments as a result of this. Table 1 displays information about the concentrations of compounds and a test matrix. 

The experimental findings were mathematically processed using the Neural Statistica 12 Network software program (Statsoft, Tulsa, OK, USA) [48].

### 2.3. Characterization of Se NPs-CTAC

The microstructure of Se NPs-CTAC were examined using a Carl Zeiss Libra 120M transmission electron magnifying lens (Carl Zeiss Microscopy, Oberkochen, Germany). Se NPs-CTAC were coupled by ultrasonic scattering of a 1:1 arrangement of test and water onto copper lattices with a carbon substrate. During transmission electron microscopy (TEM), the Libra 120M FEM thermionic weapon (Carl Zeiss Microscopy, Oberkochen, Germany) was used with a quickening voltage of 120 kV [49,50].

The amount of Se NPs was measured using Photocor-Complex gadget (Antek-97, Moscow, Russia) and dynamic light scattering (DLS) with the DynaLS software 2.0 (Photocor, Moscow, Russia). Tests of Se NPs were attenuated four times with refined water for estimation [49]. The study of the ζ-potential was carried out by electroacoustic spectroscopy using a DT-1202 setup (Dispersion Technology Inc., Lakewood, NJ, USA).

The spectral characteristics of Se NPs samples were studied by Fourier transform infrared spectroscopy (FTIR) using FSM-1201 spectrometer (Infraspec, Saint Petersburg, Russia) and UV–visible spectroscopy using SF-56 spectrometer (OKB “Spectrum”, Saint Petersburg, Russia). 

### 2.4. The Stability of Se NPs-CTAC

The pH and ionic strength of the solution were varied to assess the stability of Se NPs. To investigate the influence of pH on the stability of positive and negative Se NPs sols, solutions with varying pH levels were added to the samples in a 1:1 ratio. A solution was prepared by combining phosphoric acid, acetic acid and boric acid in a proportion of 0.04 M. In a 2 L volumetric flask, 5.49 mL H_3_PO_4_, 4.58 mL CH_3_COOH, and 4.95 g H_3_BO_3_ were combined. With distilled water, the capacity was reduced to 2 L. A 700 mL 0.2 M NaOH solution was also made separately. To create the needed pH buffer, 100 mL of acidic solution was mixed with x mL of 0.2 M NaOH solution. The resultant solution’s average hydrodynamic radius and ζ-potential were then measured [51]. The volume of NaOH utilized at the various pH levels was measured and is shown in Appendix A.

To study the effect of the ionic strength of solutions on the stability of Se NPs, five series of solutions were created. The particle radius was measured in solutions of sodium chloride (NaCl), ferric chloride (FeCl_3_), barium chloride (BaCl_2_), sodium sulfate (Na_2_SO_4_), and potassium phosphate (K_3_PO_4_) at 0.1 M, 0.25 M, 0.5 M, 0.75 M, and 1 M concentrations. Se NP sol and salt solution samples were combined in a 1:1 volume ratio [52].

To study the effect of storage time on the stability of Se NPs, samples were examined by DLS for 14 days. This study was conducted every two days. The samples were stored in the refrigerator at 0–+4 °C.

### 2.5. Potential Antibacterial and Fungicidal Activity of Se NPs-CTAC 

*Escherichia coli* and *Micrococcus luteus*, as well as *Mucor*, were cultivated to study the inhibitory properties of the precursor (selenious acid), stabilizer (CTAC), reducing agent (ascorbic acid), and Se NPs-CTAC. 

The main stages of the preparation of nutrient media and the cultivation of microorganisms are described in detail in Appendix A. Table 2 shows how were prepared samples of each preparation. 

For inoculation, a standard microbial suspension was used. To prepare a microbial suspension, microorganisms were washed off from the nutrient medium with distilled water. The resulting suspension was filtered through a sterile cotton-gauze filter and diluted with distilled water to a concentration equivalent to 0.5 according to the McFarland standard [53], diluted 100 times in isotonic solution. McFarland standard was confirmed via optical density of suspensions using Shimadzu UV-1800 spectrophotometer (Shimadzu, Tokyo, Japan). Absorption at a wavelength of 625 nm was in the range of 0.08–0.10 which corresponds to concentration of 1.5 × 10^8^ CFU/mL for bacteria and 0.75 × 10^9^ CFU/mL for *Mucor* [53]. The final concentration of microorganisms in solution after dilutions was 3 × 10^5^ CFU/mL for *Escherichia coli* and *Micrococcus luteus*, and 1.5 × 10^5^ CFU/mL for *Mucor*. Inoculum of 0.1 mL was added to test tubes containing 0.1 mL of samples 1–5 of selenious acid, ascorbic acid, CTAC or Se NPs-CTAC, according to Table 2. Resulting suspensions in the amount of 100 µL were sown on the nutrient medium. Petri dishes were incubated at +37 °C for 24 h (*Escherichia coli* and *Micrococcus luteus*) and 48 h (*Mucor*) in a thermostat (Binder GmbH, Tuttlingen, Germany). After incubation, CFU were calculated by the method of serial dilutions. For this, 9 mL of distilled water were added to test tubes. A volume of 1 mL of inoculate was added to the first tube, then it was sequentially transferred to the next tube until the inoculate was diluted 10^5^ times [54]. The procedure was repeated for each of sample. The experiment was performed in three repetitions.

The potetnial antibacterial activity of Se NPs was also studied by the disk diffusion method with determination of diameter of inhibition zones in mm [55]. *Escherichia coli* and *Micrococcus luteus* were also used for the potetnial antibacterial activity assay. Briefly, sterile filter paper discs (2 mm) were impregnated with samples 1–5 of selenious acid, ascorbic acid, CTAC or Se NPs-CTAC, according to Table 2. Petri dishes were filled out by the nutrient media prepared by the same method (Appendix A). The surface of the nutrient media was inoculated by bacteria suspension using Drygalsky spatula [56]. Finally, the impregnated disks were placed on the inoculated nutrient media and incubated at 37 °C for 24 h. After incubation, the diameter of the growth inhibition zones was measured. The tests were triplicated.

### 2.6. Statistical Data Processing

Experimental data processing was carried out in the Statistica 12.0 software (Statsoft, Tulsa, OK, USA). The Neural Statistica Network package (Statsoft, Tulsa, USA) was used to construct ternary surfaces. Graphical processing of experimental data was carried out in the Origin Pro software (https://www.originlab.com/, OriginLab, Northampton, MA, USA).

## 3. Results and Discussion

### 3.1. Optimization of Parameters for the Synthesis of Se NPs-CTAC

Table 3 displays the numerical values of the average hydrodynamic radii of particles and ζ-potential of samples 1–9 prepared according to Table 1. 

The determination of the concentration of Se NPs is described in Appendix A. Figure 2 depicts the histogram of the hydrodynamic radii distribution of Se NPs in Sample 1.

The DLS findings revealed that all samples exhibited a monomodal particle size distribution. The lowest size (r_av_ < 22 nm) of Se NPs was discovered in samples 1, 2, 4, and 8. It is crucial to note that samples 1, 2, 3, 4, 5, 7, and 9 coagulated within 24 h after synthesis. As a result, samples 6 and 8 were chosen to explore the influence of ionic strength and pH on the stability of Se NPs sols. It is worth noting that the negative value of the electrokinetic potential in samples 3, 6 and 9 is due to the fact that desorption of the positive layer of CTAC occurred due to the achievement of the second critical micelle concentration [47]. As a result, the potential-forming layer is formed due to negatively charged ions of selenious acid.

Three-dimensional ternary dependencies were created as a consequence of mathematical processing of the data (Figure 3 and Figure 4).

The quantities of selenious acid and CTAC have the largest influence on the average hydrodynamic radius, according to an examination of the ternary surface (Figure 3). At 0.062–0.178 mol selenious acid and 0.03 mol CTAC, the minimal size of Se NPs was obtained. When the quantity of selenious acid was less than 0.06 mol or greater than 0.18 mol, as well as when the amount of CTAC was greater than 0.04 mol, the maximum size of Se NPs was generated.

### 3.2. Transmission Electron Microscopy of Se NPs-CTAC

Interpretation of results of TEM is schematically presented in Figure 4.

The ζ-potential of Se NPs was discovered to be strongly dependent on the concentrations of CTAC and ascorbic acid (Figure 4). The surface of the Se NPs develops a negative charge (ζ > 0) when exposed to >0.06 mol CTAC and 1 mol ascorbic acid, and a positive charge (ζ > 0) when exposed to >0.06 mol CTAC and 1 mol ascorbic acid. It was also discovered that a surface potential-forming layer, comprised of acidic residues of selenious acid (SeO_3_^2−^) and CTAC, forms on the surface of positive and negative Se NPs, determining the particle charge. Figure 4 depicts the relationship between the appearance of the micelles and the component ratio. 

TEM was used to confirm the assumption regarding the structure of the micelles of positive and negative Se NPs. TEM images (Figure 4) revealed that negative Se NPs contained one percent of 50–85 nm spherical particles, which corresponds with results presented in [49]. One percent with a diameter of 20–50 nm was also identified in positive Se NPs samples. 

### 3.3. Computer Quantum Chemical Modeling of CTAC Molecule and Se NPs-CTAC Molecular Complex

Computer quantum chemical modeling was used to simulate the CTAC molecule and Se NPs-CTAC molecular complex (Figure 5, Table 4).

Quantum chemical computer modeling results show that the total energy of the CTAC molecule is −803.904 kcal/mol, and the energy of the Se-CTAC molecular complex is −12,795.691 kcal/mol. A decrease in the total energy of the system indicates the energetic advantage of chemical bond formation between Se and CTAC [57].

### 3.4. Spectral Characteristics of Se NPs-CTAC 

The results of the study of the spectral characteristics of Se NPs-CTAC are shown in Figure 6.

The analysis of the FTIR spectrum of CTAC (Figure 6a) showed the presence of oscillation bands at 1377 and 1651 cm^−1^, which correspond to symmetric oscillations of the CH_2_ group, as well as bands at 2851, 2870, 2887, 2899, 2916, 2920 and 2924 cm^−1^, characteristic of the CH_2_ group. In the range from 1265 to 1344 cm^−1^, bands of symmetrical oscillations of the CH_3_ group are observed. The oscillation bands at 2951, 2955, 2959, 3021 and 3028 cm^−1^ are also characteristic of the CH_3_ group. In addition, oscillation bands at 1479 and 1494 cm^−1^ are characteristic of the NH^+^ bond [58].

The analysis of the FTIR spectrum of Se NPs-CTAC (Figure 6a) showed fluctuations characteristic of the CH_2_ group were also observed at 1377, 2851 and 2920 cm^−1^. Bands corresponding to fluctuations of the CH_3_ group are observed at 2951, 3026 cm^−1^ and in the range from 1265 and 1344 cm^−1^. Vibrations of the CH group (3399 and 3483 cm^−1^) and NH^+^ bonds (1479, 1493 cm^−1^) are also present on the FTIR spectrum. In addition, the FTIR spectrum of Se NPs-CTAC contains fluctuations at 592 and 665 cm^−1^, characteristic of Se, and fluctuations at 721 cm^−1^, characteristic of the Se-O bond [59].

As a result of the analysis of the FTIR spectra, it can be concluded that there is a decrease in the intensity of bands in the regions from 2851 to 3026 cm^−1^ and from 1265 to 1377 cm^−1^, characteristic of the CH_2_ and CH_3_ groups, which indicates that CTAC was present on the surface of Se NPs, oriented by a hydrophobic center to Se NPs, which is consistent with the results of computer quantum chemical modeling.

Analysis of the obtained absorption spectrum of UV–visible spectroscopy (Figure 6b) showed that Se NPs-CTAC have an absorption band at λ = 335 nm. Thus, the obtained spectral characteristics correspond to the results presented in other works [47,60,61].

### 3.5. The Stability of Se NPs-CTAC at Different pH

In the next step, a study was conducted on the effect of pH on the stability of Se NPs. Figure 7 shows the pH dependence of ζ-potential and the mean hydrodynamic radius for positive and negative Se NP sols.

The pH of negative Se NPs sols and positive Se NPs sol’s reversal show fundamentally different dependences on the average hydrodynamic radius of nanoparticles, which confirms the results obtained in the previous work [47]. It was found that at pH 10, a large loss of aggregation stability of the positive Se NP sol was observed, which is confirmed by the decrease in ζ-potential at these pH values. At 5 < pH < 8, an increase in particle size of the positive Se NP sol was again observed, probably because the isoelectric point of CTAC was reached [57]. For negative Se NPs sols, increasing pH does not affect aggregate stability, indicating high stability across the studied range [62,63].

### 3.6. The Stability of Positive and Negative Se NPs Sols at Various Ions

In the next step, the influence of various ions on the stability of positive and negative Se NPs sols was studied. The effects of the presence of cations and anions on the stability of Se NPs were analyzed by DLS. The obtained dependence of the mean hydrodynamic radius of Se NPs on the concentration of cations and anions is shown in Figure 8.

An analysis of the size dependence of Se NPs (Figure 8) showed that anions had a slight effect on particle aggregation in sols containing negative Se NPs, and particle size did not change over the entire concentration range [64]. It can be seen that Na^+^ ions have no effect on the size of the nanoparticles. At the same time, according to the data in Figure 8, Ba^2+^ and Fe^3+^ ions have the greatest aggregation effect on the negative Se NPs sol. The same results were obtained in study of Se NPs stabilized with cocamidopropyl betaine [47]. Ba^2+^ ions increased the particle size of the negative Se NPs sol from 25 nm to 175 nm, accompanied by precipitate formation [65]. The lowest concentration of Fe^3+^ ions (0.1 mol/L) resulted in a slight increase in particle size to 30 nm, while higher concentrations (up to 1 mol/L) resulted in a sharp increase to 225 nm.

For the positive Se NP sol, an inverse relationship was found. Anions have a significant effect on the aggregation of Se NPs. This is because PO_4_^3−^ and SO_4_^2−^ ions cause an increase in the ionic strength of the solution, leading to significant expansion of Se NPs up to 200 and 100 nm, respectively [26,66,67]. The SO_4_^2−^ ion dependence analysis showed that increasing the concentration of the Na_2_SO_4_ solution to 0.25 M did not change the radius of the Se NPs and the solution remained stable. A further increase in the SO_4_^2−^ ion concentration significantly increases the mean hydrodynamic radius of Se NPs. In the following dependencies, the initial solidification process of Se NPs resulted in the precipitation of a highly solidifying electrolyte (Na_3_PO_4_) over the considered concentration range [65]. At concentration 1 mol/L PO_4_^3−^ Se NPs had the largest size and solidification rate.

Figure 8 shows that cations do not significantly affect the CTAC-stabilized positive Se NP sol. It is worth noting that Fe^3+^ ions influence the size of the positive Se NP sol. This is because there are three Cl^−^ anions for every Fe^3+^ cation, which have a corresponding effect on the size of Se NPs. Since Se NPs are used in combination with various impurities in veterinary, pharmaceutical, cosmetic compositions and even biologically active substances, the data obtained consider the high potential of practical application of Se NPs-CTAC [60,61,67,68,69,70].

### 3.7. The Stability of Se NPs-CTAC at Storing

At the next stage, the stability of the positive and negative Se NPs sols at the exposure time was investigated. The data obtained are presented in Figure 9.

Figure 9 shows that the average hydrodynamic radius of the positive Se NPs sol increased significantly during the storage of samples. During 14 days of the experiment, the average hydrodynamic radius of Se NPs increased from 25 ± 3 to 110 ± 9 nm, which is quite normal for Se NPs complexes [11,71,72]. At the same time, it is important to note that the average hydrodynamic radius of the negative Se NPs sol did not change significantly during the experiment (total difference was 6 nm) which characterizes its high stability at storing. 

### 3.8. Potential Antimicrobial Activity of Se NPs-CTAC

The potential antimicrobial activity of preparations containing precursors, stabilizers, reducers and Se NPs-CTAC (Table 2) was studied against *Escherichia coli* and *Micrococcus luteus*, as well as *Mucor*. The results are shown in Table 5.

According to Table 5, ascorbic acid as well as selenic acid have not antimicrobial activity. Samples of these series were characterized by the growth of numerous colonies reaching 300 CFU/mL after dilutions, which corresponds to results of other works [73,74]. The results of the experiment revealed potential antimicrobial activity in the CTAC series. The minimum inhibitory concentration (MIC) was 8.322 mmol/L for *Escherichia coli* and *Micrococcus luteus*, and 83.221 mmol/L for *Mucor*. Similar results were reported in studies of cetyltrimethylammonium chloride as well as cetyltrimethylammonium bromide [75,76,77]. Expected, Se NPs-CTAC samples had the highest activity. The presence of 0.1940 mmoll/L Se NPs-CTAC provided inactivation of *Escherichia coli*, *Micrococcus luteus* and *Mucor.* Interestingly, MIC for *Micrococcus luteus* was detected at 0.0194 mL/L Se NPs-CTAC. Thus, potential antibacterial activity of Se NPs-CTAC against *Micrococcus luteus* is higher than other forms of nanoparticles reported in [78,79,80,81]. MIC of Se NPs-CTAC for *Escherichia coli* (0.1940 mmoll/L) was lower than MIC of Se NPs stabilized with polysorbate [82], bovine serum albumin and chitosan [11], as well as green synthesized Se NPs [83]. Regarding potential antifungal activity, the results obtained corresponds to results of other authors tested various forms and complexes of Se NPs on *Mucor*, *Penicillium digitatum*, *Rhizoctonia solani*, *Lichtheimia corymbifera* and *Syncephalastrum racemosum* [84,85,86]. 

High potential of antibacterial activity of Se NPs-CTAC was also confirmed using the disk diffusion method. The data obtained are presented in Figure 10 and Table 6.

Analysis of Figure 10a showed that in samples of ascorbic and selenic acids, *Micrococcus luteus* culture growth to the edges of the disks was observed. There was no inhibition zone. A stable isolate was observed in samples 1–3 of CTAC where the inhibition zones were 17–31 mm (Table 6). In all samples of Se NPs-CTAC, a stable isolate was observed with an inhibition zones 18–31 mm. According to Table 6, the largest inhibition zones were observed in both sample of CTAC (832.21 mmol/L) and Se NPs-CTAC (19.39 mmol/L). Similarly, Figure 10b shows that ascorbic and selenic acids have not an antibacterial effect against *Escherichia coli*—all samples were characterized by the continuous growth of disks without inhibition zones. The greatest effect was observed in sample of Se NPs-CTAC with concentration 19.39 mmol/L (33 mm). The results obtained corresponds to results of other authors observed inhibition zones of 12–25 mm at treatment of bacteria by Se NPs [87]. In addition, Islam et al. [88] described inhibition zones of 6–23 mm observed at treatment of *Aspergillus niger* with biosynthesized Se NPs, which also confirms potential antifungal activity of Se NPs. 

The cationic surfactant CTAC belongs to a class of quaternary ammonium compounds known to have antimicrobial properties. These compounds actively act on sensitive bacterial cells, adsorb to and penetrate cell wall components [75]. Jyoti et al. [89] in their study demonstrated a mechanism of fungicidal action of CTAC based on cellular and nuclear membrane damage. After interacting with the constituents of the microbial cell membrane, disruptions occur in the molecular matter system, leading to the degradation of heterofunctional organic molecules. Consequently, the formation of micelles from quaternary ammonium compounds and cell membrane components ultimately leads to membrane solubilization and cell lysis [90,91]. Significant antibacterial and antifungal effects were observed in CTAC-treated samples in all cultures, and these results are consistent with those of other researchers [92,93]. Se NPs have natural antibacterial and fungicidal effects on various crops [94,95,96,97,98,99]. Researchers are now observing and explaining different mechanisms of action of Se NPs on microorganisms. According to Han et al. [100], the bactericidal activity of Se NPs is based on the degradation of microbial proteins. Lesnichaya et al. [101] reported that SeNPs contribute to the inactivation of the natural mechanisms of membrane transport of ions and nutrients across the cell wall, blocking the vital activity of microbial cells. Developing studies on the mechanism of antibacterial activity of Se NPs, Lesnichaya et al. [102] discovered that the slow releasing Se ions from the surface of Se NPs interacts with -SH, -NH, or -COOH functional groups of proteins and enzymes, and subsequently can lead to the loss of their tertiary and quaternary structure and function. The authors also mentioned that Se NPs inhibit the activity of dehydrogenase enzymes and disrupt the integrity of cell membranes. Se NPs is known to cause overproduction of reactive oxygen species, disruption of membrane potential, and degradation of internal adenosine triphosphate [103,104]. Liang et al. [105] found that Se NPs inhibit the ability of bacteria to adhere to surfaces and form bacterial membranes. An equally interesting effect worth noting is the photocatalytic activity of Se NPs against bacteria reported by Sahoo et al. [106]. Taking into account the results of other researchers and the results of this study, the hypothesis of antibacterial action of CTAC, Se NPs and Se NPs-CTAC was proposed (Figure 11).

The presented hypothesis can be summarized as follows: introduction of the Se NPs-CTAC molecular complex into bacterial or fungal cell cultures leads to degradation of proteins and polysaccharides and disruption of microbial cell structure, allowing an enhanced penetration of Se NPs via damaged cytoplasmic membrane, which causes oxidative damage and changes in the intensity of reactive oxygen species [107,108]. Further study of Se NPs-CTAC will be aimed at confirming this hypothesis.

Therefore, due to the synergistic effect of its components, the Se NPs-CTAC molecular complex has potential antibacterial and antifungal properties, which determines the broad potential of this formulation for practical application.

## 4. Conclusions

In this work, a method for the synthesis of Se NPs-CTAC was developed. The parameters for the synthesis of Se NPs-CTAC were studied and optimized.

According to the results obtained, Se NPs in stable sols have a radius of 21–27 nm. It was found that the ζ-potential of Se NPs is significantly affected by the concentrations of CTAC and ascorbic acid. The surface of Se NPs acquires a negative charge (ζ > 0) when exposed to >0.06 mol CTAC and 1 mol ascorbic acid and a positive charge (ζ > 0) when exposed to >0.06 mol CTAC and 1 mol ascorbic acid.

It was found that a surface potential formation layer consisting of acidic selenite residues (SeO_3_^2-^) and CTAC is formed on the surface of positive and negative Se NPs sols and determines the charge of the particles. A study of the samples using TEM showed that both negative and positive Se NPs sols contain a fraction of spherical particles. The particle size in negative Se NPs sol was 50–85 nm; in positive Se NPs sol—20–50 nm. According to the results of quantum chemical computer modeling, the total energy of the CTAC molecule is −803.904 kcal/mol and the energy of the Se NPs-CTAC molecular complex is −12,795.691 kcal/mol. The decrease in the total energy of the system indicates that the formation of a chemical bond between Se NPs and CTAC is energetically favorable. The stability of the Se NP-CTAC complex was studied at different pH values and in the presence of different ions. It was found that the negative Se NPs sol was the most stable. 

It was found that Se NPs-CTAC have potential antibacterial and antifungal activities compared to individual solutions of precursor, stabilizer, and reducing agent. Experimental results show that the minimum concentrations of Se NPs-CTAC required to exhibit potential antibacterial properties are 0.194 mmol/L for *Escherichia coli*, 0.0194 mmol/L for *Micrococcus luteus* and 0.194 mmol/L for *Mucor*. Disk diffusion methods revealed inhibition zones of 12–33 mm for *Escherichia coli* and 18–31 mm for *Micrococcus luteus*, which also confirms potential antibacterial activity of Se NPs-CTAC at various concentrations. The enhanced potential antibacterial and antifungal activity of Se NPs-CTAC may be due to the synergistic effect of the antibacterial properties of CTAC and Se NPs. Further research will be conducted to study the mechanisms of influence of CTAC, Se NPs and Se NPs-CTAC on microbial cells.

The research results can be further applied in the development of products containing Se NPs, such as veterinary drugs, pharmaceuticals, and cosmetics. We also plan to synthesize Se NPs stabilized with various quaternary ammonium compounds and study their activity and potential ways of application.

## Figures and Tables

**Figure 1 nanomaterials-13-03128-f001:**
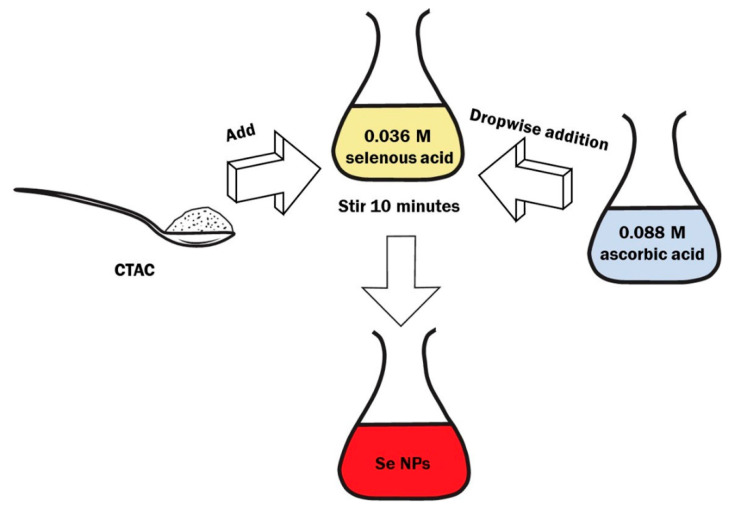
The schematic diagram of the synthesis of Se NPs-CTAC.

**Figure 2 nanomaterials-13-03128-f002:**
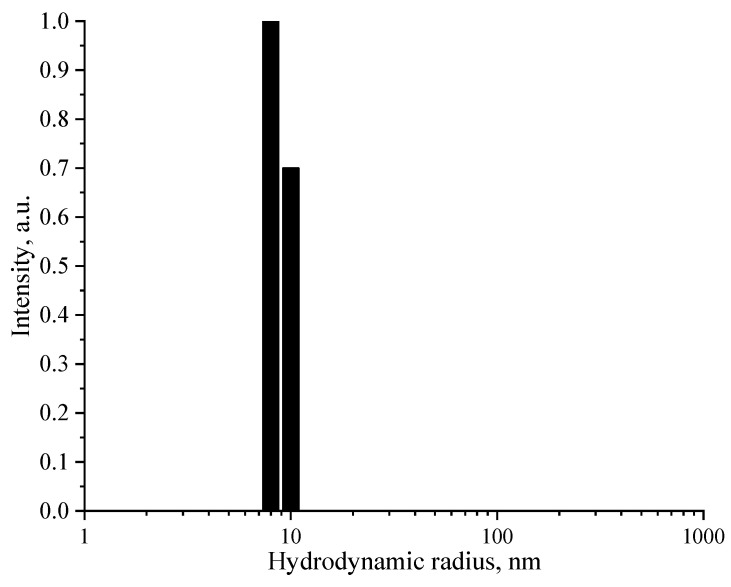
Histogram of the average hydrodynamic radius distribution of Se NPs (Sample 1).

**Figure 3 nanomaterials-13-03128-f003:**
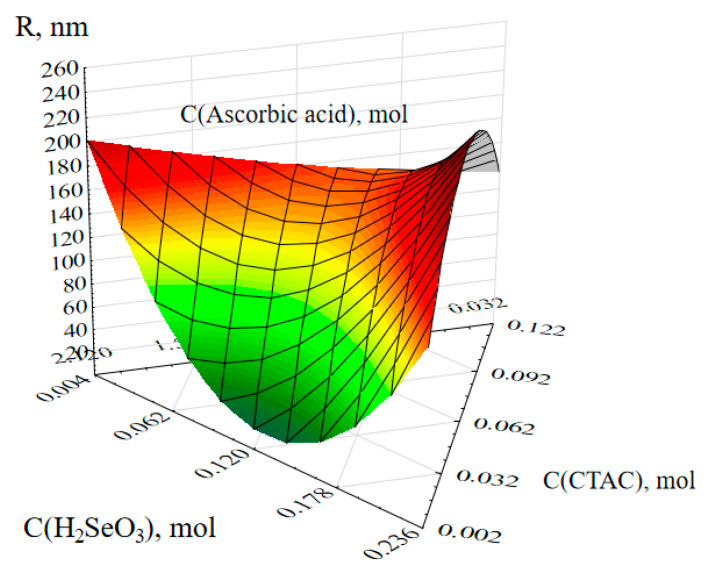
Ternary surface describing the relationship between the average hydrodynamic radius and concentrations of selenious acid and CTAC.

**Figure 4 nanomaterials-13-03128-f004:**
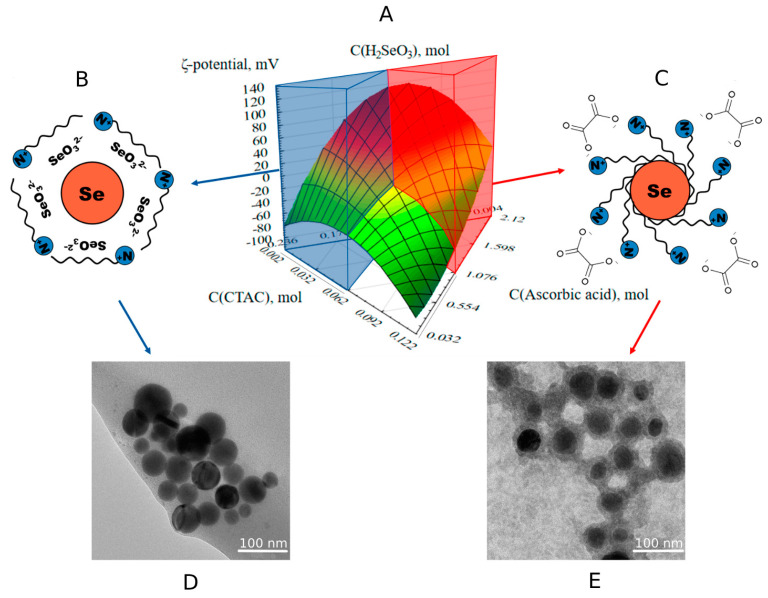
Ternary surfaces describing the relationship between the ζ-potential of Se NPs and the concentrations of selenious acid and CTAC (**A**) with schematic models of negative (**B**) and positive (**C**) micelles of Se NPs and TEM micrographs of negative (**D**) and positive (**E**) Se NPs sols.

**Figure 5 nanomaterials-13-03128-f005:**
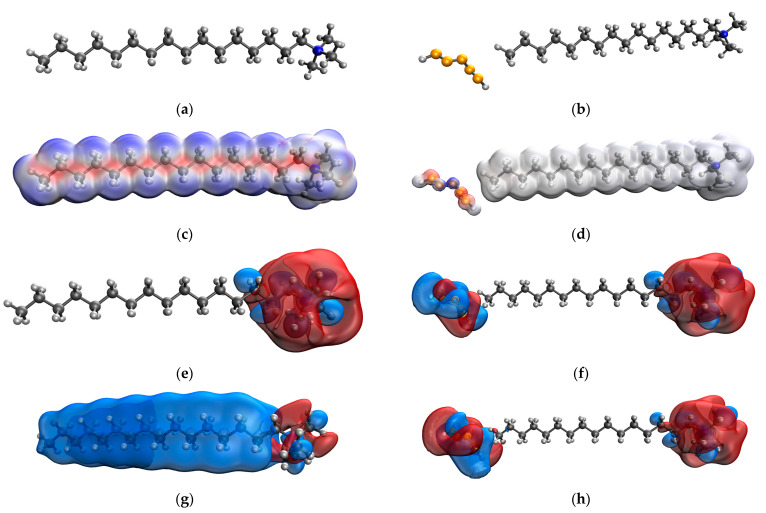
Computer quantum chemical modeling of CTAC molecule and interaction of Se NPs with CTAC molecule: molecular model of CTAC (**a**), molecular model of Se NPs-CTAC (**b**), electron density distribution of CTAC (**c**), electron density distribution of Se NPs-CTAC (**d**), highest occupied molecular orbital of CTAC (HOMO) (**e**), highest occupied molecular orbital of Se NPs-CTAC (HOMO) (**f**), lowest unoccupied molecular orbital of CTAC (LUMO) (**g**), and lowest unoccupied molecular orbital of Se NPs-CTAC (LUMO) (**h**).

**Figure 6 nanomaterials-13-03128-f006:**
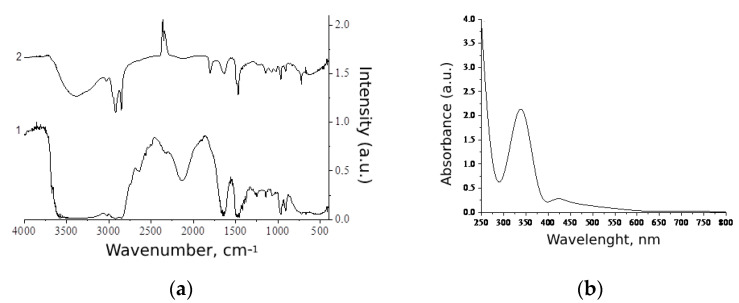
Spectral characteristics of Se NPs-CTAC: (**a**) FTIR spectra of CTAC (1) and Se NPs-CTAC (2); (**b**) absorption spectrum of Se NPs-CTAC.

**Figure 7 nanomaterials-13-03128-f007:**
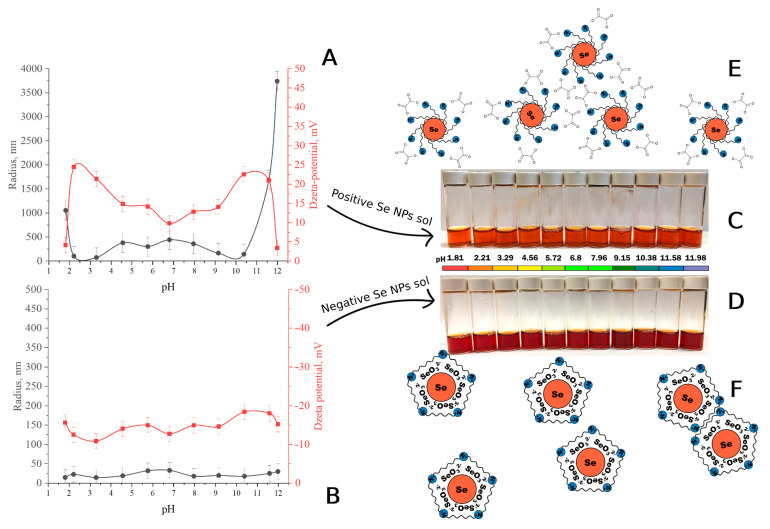
Stability of Se NPs-CTAC at different pH: dependences of the ζ-potential and the average hydrodynamic radius of positive (**A**) and negative (**B**) Se NPs sols on pH; appearance of samples of positive (**C**) and negative (**D**) Se NPs sols at different pH; schematic models of positive (**E**) and negative (**F**) Se NPs sols.

**Figure 8 nanomaterials-13-03128-f008:**
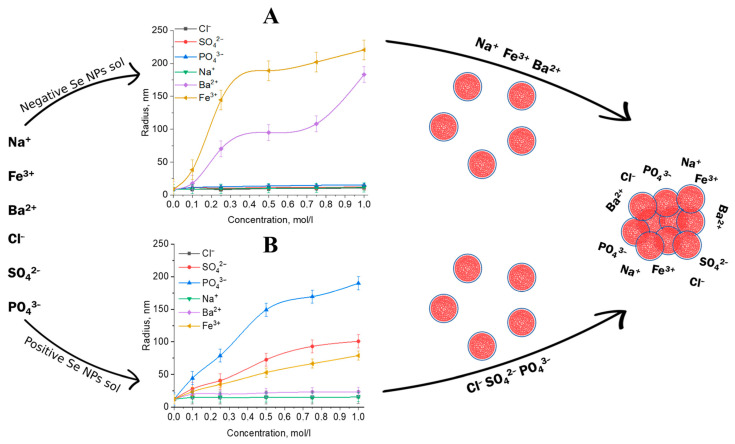
Dependence of the average hydrodynamic radius of particles in negative (**A**) and positive (**B**) Se NPs sols on the concentration of various ions.

**Figure 9 nanomaterials-13-03128-f009:**
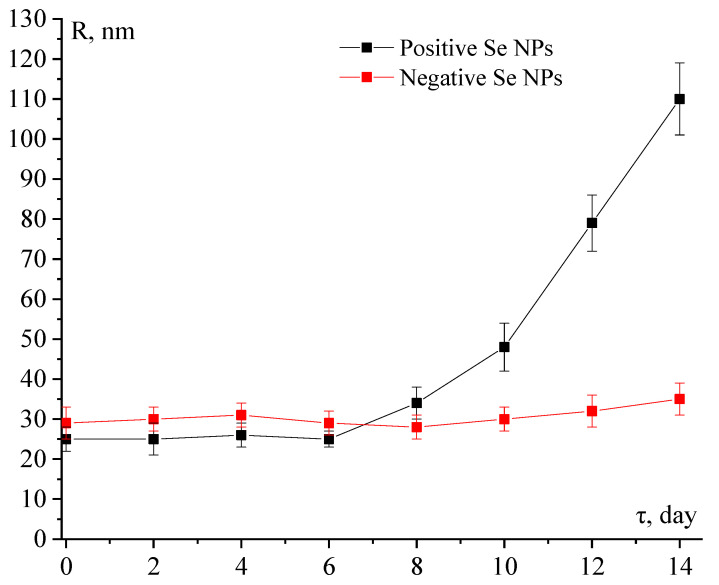
Dependence of the average hydrodynamic radius of particles (R, nm) on the exposure time (τ, day).

**Figure 10 nanomaterials-13-03128-f010:**
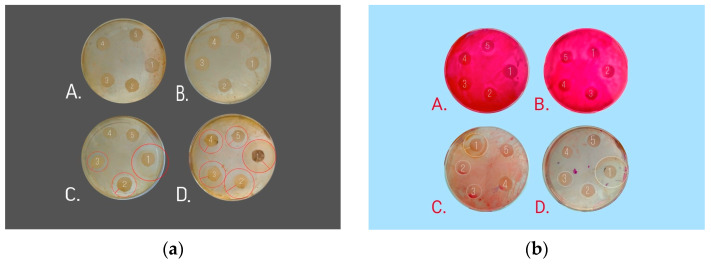
Study of potential antibacterial activity of Se NPs-CTAC by the disk diffusion method: (**a**)—*Micrococcus luteus*; (**b**)—*Escherichia coli*; A—selenic acid, B—ascorbic acid; C—CTAC; D—Se NPs-CTAC.

**Figure 11 nanomaterials-13-03128-f011:**
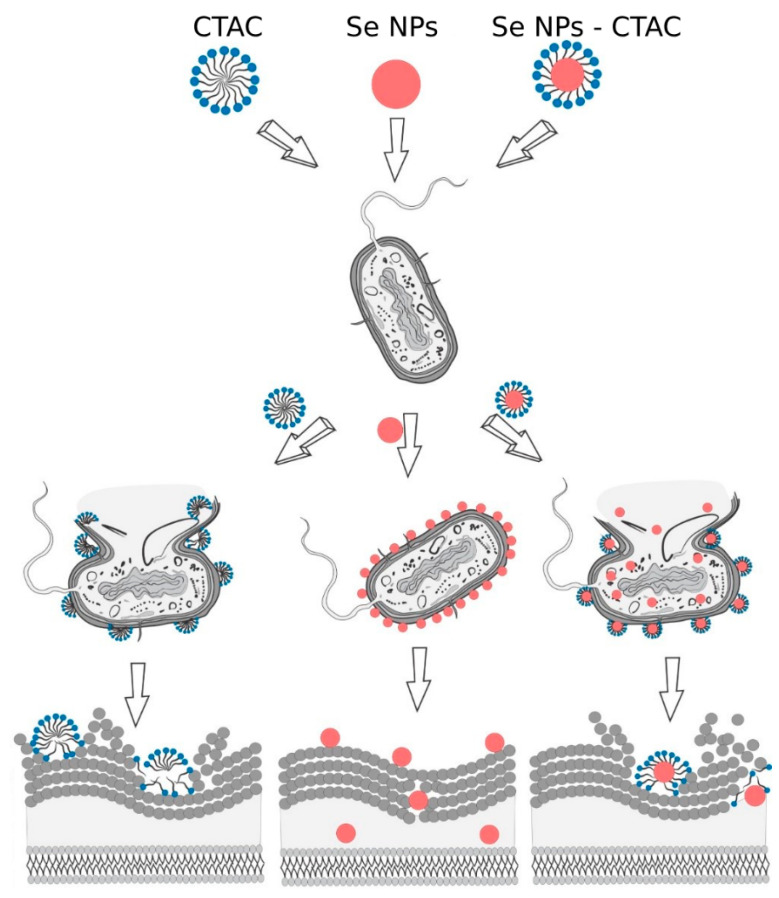
The scheme of action of CTAC, Se NPs and Se NPs-CTAC on microbial cell.

**Table 1 nanomaterials-13-03128-t001:** Variation levels of parameters and matrix of the experiment.

Sample Num.	C (H_2_SeO_3_) (mg/mL)	C (CTAC) (mg/mL)	C (Ascorbic Acid) (mg/mL)
1	0.48	0.61	5.83
2	0.48	4.86	46.6
3	0.48	38.89	372.8
4	3.8	0.61	46.6
5	3.8	4.86	372.8
6	3.8	38.89	5.83
7	30.4	0.61	372.8
8	30.4	4.86	5.83
9	30.4	38.89	46.6

**Table 2 nanomaterials-13-03128-t002:** Samples for the study of potential antimicrobial properties.

Sample	Concentration, mmol/L				
	1	2	3	4	5
Selenious acid	43.9300	4.3930	0.4393	0.0439	0.0044
Ascorbic acid	43.7500	4.3750	0.4375	0.0438	0.0044
CTAC	832.2100	83.2210	8.3220	0.8320	0.0832
Se NPs-CTAC	19.3900	1.9390	0.1940	0.0194	0.0019

**Table 3 nanomaterials-13-03128-t003:** Average hydrodynamic radii of Se NPs.

Sample Num.	ζ-Potential (mV)	r_av_ (nm)
1	28.04	21 ± 4
2	36.49	21 ± 2
3	−106.23	113 ± 15
4	52.32	21 ± 2
5	16.21	64 ± 4
6	−76.20	27 ± 3
7	15.68	28 ± 2
8	11.38	21 ± 3
9	−1.72	25 ± 3

r_av_—average hydrodynamic radius (n ± PDI).

**Table 4 nanomaterials-13-03128-t004:** Numerical values of the parameters of computer quantum chemical modeling.

Substance	Bond	E	HOMO	LUMO	η
CTAC	-	−803.904	0.022	0.084	0.031
	Se	−12,795.691	−0.005	−0.000	0.003
Se	-	−11,991.801	−0.141	−0.042	0.050

**Table 5 nanomaterials-13-03128-t005:** Potential antimicrobial activity of the precursor, stabilizer, reducing agent, and Se NPs-CTAC.

Preparation	Sample	Concentration, mmol/L	CFU, ×10^3^ in 1 mL		
			*Escherichia coli*	*Micrococcus luteus*	*Mucor*
Ascorbic acid	1	43.7500	0.26 ± 0.03	0.24 ± 0.05	0.11 ± 0.02
	2	4.3750	0.23 ± 0.04	0.26 ± 0.02	0.12 ± 0.02
	3	0.4375	0.23 ± 0.03	0.23 ± 0.04	0.11 ± 0.01
	4	0.0438	0.25 ± 0.02	0.26 ± 0.03	0.14 ± 0.01
	5	0.0044	0.27 ± 0.02	0.25 ± 0.03	0.13 ± 0.02
Selenic acid	1	43.9300	0.24 ± 0.03	0.25 ± 0.02	0.12 ± 0.03
	2	4.3930	0.25 ± 0.04	0.23 ± 0.05	0.13 ± 0.01
	3	0.4393	0.25 ± 0.02	0.22 ± 0.04	0.11 ± 0.02
	4	0.0439	0.24 ± 0.04	0.24 ± 0.05	0.12 ± 0.02
	5	0.0044	0.26 ± 0.02	0.27 ± 0.02	0.13 ± 0.01
CTAC	1	832.2100	0	0	0
	2	83.2210	0	0	0
	3	8.3220	0	0	0.02 ± 0.01
	4	0.8320	0.08 ± 0.03	0.03 ± 0.01	0.02 ± 0.01
	5	0.0832	0.11 ± 0.01	0.04 ± 0.01	0.05 ± 0.02
Se NPs-CTAC	1	19.3900	0	0	0
	2	1.9390	0	0	0
	3	0.1940	0	0	0
	4	0.0194	0.03 ± 0.01	0	0.02 ± 0.01
	5	0.0019	0.04 ± 0.01	0.03 ± 0.01	0.03 ± 0.01

**Table 6 nanomaterials-13-03128-t006:** Potential antibacterial activity of Se NPs-CTAC.

Preparation	Sample	Concentration, mmol/L	Inhibition Zone, mm	
			*Escherichia coli*	*Micrococcus luteus*
Ascorbic acid	1	43.7500	0	0
	2	4.3750	0	0
	3	0.4375	0	0
	4	0.0438	0	0
	5	0.0044	0	0
Selenic acid	1	43.9300	0	0
	2	4.3930	0	0
	3	0.4393	0	0
	4	0.0439	0	0
	5	0.0044	0	0
CTAC	1	832.2100	26 ± 3	31 ± 4
	2	83.2210	17 ± 4	20 ± 2
	3	8.3220	12 ± 2	17 ± 4
	4	0.8320	0	0
	5	0.0832	0	0
Se NPs-CTAC	1	19.3900	33 ± 2	31 ± 3
	2	1.9390	21 ± 5	26 ± 2
	3	0.1940	15 ± 2	22 ± 3
	4	0.0194	11 ± 2	20 ± 2
	5	0.0019	12 ± 4	18 ± 1

## Data Availability

All raw data are available upon request from the corresponding author.

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
