# Peer review of "Synthesis, Characterization and Potential Antimicrobial Activity of Selenium Nanoparticles Stabilized with Cetyltrimethylammonium Chloride"

_nanomaterials, 2023, doi:10.3390/nano13243128_

Round 1

Reviewer 1 Report

Comments and Suggestions for Authors

This manuscript mainly focused on the synthesis and antimicrobial activity of SeNPs stabilized with cethyltrimethylammonium chloride. The manuscript needs significant improvement before acceptance for publication. My detailed comments are as follows:

1. All the data presented in Table 1 and Table 2 should keep the same significant figures.

Why the concentration of H2SeO3 in Table 1 (0.48 mg/ml) and Table 2 (0.475 mg/ml) is different?

2. Please explain why SeNPs possess negative ζ-potential under the highest concentration of CTAC at 38.89 mg/ml.

3. The PDI values of different samples should be added in Table 5.

4. The storage stability of SeNPs should also be investigated.

5. The interaction between the CTAC and SeNPs should be confirmed through several spectral characterization, refer to the reference (Antioxidants, 2022, 11, 240).

6. In table 4, how to determine the concentration of SeNPs, the author should explain. 

7. What is “Dzeta” in ordinate in Fig. 7 mean?

Comments on the Quality of English Language

The quality of English needs improving. Most sentences contain grammatical and/or spelling mistakes or are not complete sentences, especially in abstract.

The upper and lower labels of anions should be checked. For example, Line 294-309.

Line 414-416: The two sentences express the same means  

Author Response

Dear Reviewer 1,

We are grateful for your positive evaluation and for the time devoted to review our manuscript. All comments were useful and pleased us with the high level of understanding of the topic. We have addressed all recommendations as requested. All changes in the manuscript are marked by green. Please see the point-by-point answers below

  1. All the data presented in Table 1 and Table 2 should keep the same significant figures.

Tables are revised

Why the concentration of H2SeO3 in Table 1 (0.48 mg/ml) and Table 2 (0.475 mg/ml) is different?

Thank you for attentiveness, this mistake was fixed and corrected

  1. Please explain why SeNPs possess negative ζ-potential under the highest concentration of CTAC at 38.89 mg/ml.

The negative value of the electrokinetic potential is due to the fact that desorption of the positive layer of cetyltrimethylammonium chloride occurred due to the achievement of the second critical micelle concentration. As a result, the potential-forming layer is formed due to negatively charged ions of selenious acid.

  1. The PDI values of different samples should be added in Table 5.

The PDI values of different samples were added in Table 5

  1. The storage stability of SeNPs should also be investigated.

Thank you for recommendation. We carried out additional experiments and added requested data

  1. The interaction between the CTAC and SeNPs should be confirmed through several spectral characterization, refer to the reference (Antioxidants, 2022, 11, 240).

Thank you for recommendation. We added IR spectroscopy and optical spectroscopy data. Suggested references were considered.

  1. In table 4, how to determine the concentration of SeNPs, the author should explain.

The concentration of Se NPs was calculated theoretically. For this purpose, preliminary experiments were carried out, as a result of which the reaction yield (η, %) was determined.

η = m practical (Se) / m theoretical (Se),                         (1)

Thus, the molar concentration of Se NPs was calculated using the formula:

CM (Se) = m practical (Se) * V * M,                                    (2)

where CM (Se) – molar concentration, m theoretical (Se) – mass of substance, М – molar mass, V – volume of solution.

Based on formula (1):

m practical (Se) = m theoretical (Se) * η                (3)

Then:

CM (Se) = m practical (Se) * V * M = m theoretical (Se) * V * M * η, (4)

Taking into account the reaction equation for the production of Se NPs, given in the text of the article, we obtain that:

m sample (H2SeO3) / M (H2SeO3) = m theoretical (Se) / M (Se),      (5)

in this connection:

m theoretical (Se) = m sample (H2SeO3) * M (Se) / M (H2SeO3)     (6)

Based in formula (4):

CM (Se) = m practical (Se) * V * M = m theoretical (Se) * V * M * η =

m sample (H2SeO3) * M (Se) * V * M * η / M (H2SeO3)

This information was added to Supplementary (Section S1).

  1. What is “Dzeta” in ordinate in Fig. 7 mean?

Replaced by the symbol «ζ».

Tha manuscript was also checked for English quality. Thank you for suggestion.

Reviewer 2 Report

Comments and Suggestions for Authors

The synthesis and application of new nanoparticles are currently a research hotspot. This article synthesized nano selenium and studied its antibacterial properties, which is very meaningful. I suggest publishing the article after modification.

1, There are too many figures and tables in the main text. It is recommended to include some unimportant figures and tables in the supporting information.

2, The format of the tables in the article is incorrect. Some are not three line tables, while others are missing the bottom line.

3, Many references on antibacterial properties of nanomaterials have not been fully discussed and cited.

Safety Assessment of Nanomaterials for Antimicrobial Applications. CHEMICAL RESEARCH IN TOXICOLOGY, 33(5):1082-1109.

Tailoring metal-organic frameworks-based nanozymes for bacterial theranostics. BIOMATERIALS, 275:120951

Author Response

Dear Reviewer 2,

We are grateful for your positive evaluation and for the time devoted to review our manuscript. All comments were useful and pleased us with the high level of understanding of the topic. We have addressed all recommendations as requested. All changes in the manuscript are marked by green. Please see the point-by-point answers below

  1. There are too many figures and tables in the main text. It is recommended to include some unimportant figures and tables in the supporting information.

Thank you for your recommendation. In accordance with comments of other reviewers, we had to catty out additional experiments and add new figures and tables. Nevertheless, we combined tables 1 and 2 and figures 6 and 7 (new figures), but further presented figures and tables are necessary for a complete understanding and description of the essence of the entire article.

  1. The format of the tables in the article is incorrect. Some are not three line tables, while others are missing the bottom line.

The format of all tables was revised.

  1. Many references on antibacterial properties of nanomaterials have not been fully discussed and cited.

Safety Assessment of Nanomaterials for Antimicrobial Applications. CHEMICAL RESEARCH IN TOXICOLOGY, 33(5):1082-1109.

Tailoring metal-organic frameworks-based nanozymes for bacterial theranostics. BIOMATERIALS, 275:120951

Thank you for recommendation. In the introduction, a paragraph about antibacterial activity of nanomaterials was added. Suggested articles were considered.  

Reviewer 3 Report

Comments and Suggestions for Authors

            We thank the authors for their valuable contribution to this manuscript. However, the authors should pay attention to all the comments and suggestions very carefully from the reviewer and address each one by one. It will shape this manuscript into a better one. The authors are required to make major revisions to this manuscript according to the following comments and suggestions.

Title: Please correct the spelling “cethyltrimethylammonium

Abstract

(1)   Introductory sentences lack in the abstract portion. Please include them.

(2)   Line 27,37- Grammar errors, please revise

(3)   Line 39 – Shorten the sentence

(4)   Line 40 – Specify the broad applicability

(5)   The abstract portion exceeds the word count (approximately 200), please revise.

Introduction

(1)   Line 47- What are the severe issues of “Se insufficiency”? Address them.

(2)   Line 50- What is meant by “other forms”, revise the sentence.

(3)   After line 50, the authors jumped into the synthesis part, Please address the research question in detail before that.

(4)   Line 67- Abbreviate QAS

(5)   Line 69- Incomplete sentence, revise

(6)   Overall, the introduction part is not sufficient, it needs to be more descriptive and supported with recent references. And it should present the novelty of the research. Please revise.

Method

(1)   Lines 87,90,133, 157 167- Avoid using terms such as “First, Next, Then” in scientific writing. This applies to the whole manuscript.

(2)   Why did you use simple present tense? *Methodology section should be written in past passive tense. Line 87, 92,93,120 Revise

(3)   Line 98, 223, 250- Revise all the figure titles.

Ex: Please write Figure 1. instead of “Fig.1”

(4)   Line 107- Confirm if the name of the software appears correctly in the text.

(5)   The methodology should contain a separate section to present/say about the precise statistical analysis performed in the study if applicable.

(6)   Table 1- What do you mean by “Variation levels of parameters”? Does that mean different concentrations? Please revise titles.

(7)   Table 3 – Why don’t the pH values have standard deviations? Not taking replicate values is a serious mistake. Please re-do and revise

(8)   Line 118-Avoid repetition, revise

(9)   It seems like the authors have not characterized the Se NPs stabilized by CTAC using proper methods. It is very important in introducing a novel nano compound. Please perform the following and any other techniques in the literature:

UV-visible spectroscopy

Fourier-transform infrared spectroscopy (FTIR)

EDX spectrum

Zeta potential analysis

(10)      Line 133-139 Correct the chemical formulas (subscript). This applies to the whole document

(11)      Line 137- What is x ml here? Give a value or interpret here

(12)      Line 143 – Avoid using terms such as “we” and “I” in scientific writing

(13)      Line 149, 177- Revise the sentence. You do not have to write the word bacteria or mold at the end of the strain name.

(14)      How did you sterilize the medium? Filter sterilization or autoclave? Sentence is confusing, Please revise

(15)      Line 166- Culture media composition better to be included in a supplementary document

(16)      Line 172-175– These are very basic microbiological steps. Too much information, Please simplify the sentences, Revise

(17)      Line 184-193- Very basic microbiological steps, No need to elaborate this much, it is unnecessary, Please shorten the sentences.

(18)      Line 195-196 Give the incubation times for bacteria and mold separately.

(19)      Section 2.5 -Antibacterial and fungicidal activity of Se NPs needs to be thoroughly revised. The flow is confusing. Please divide it into segments and make it neat and more understandable.

Ex: media preparation, cultivation of bacteria and fungi (mold) and etc. Try to avoid elaborating on very basic microbiological steps. It interferes with the main focus of that section

(20)      The main focus of how you tested the compound (Se NP) for antimicrobial activity is not clear at all. Failing to illustrate this is a major drawback. Please specify the method. (whether it is disk diffusion, pour-plate, spread-plate or streak?)

(21)      Table 4 – The data is not clearly presented. What is 1,2,3,4,5?

Results

(1)   Results are very poorly organized and illustrated. It appears that all the results are just listed without proper order. At least, please give subtitles/subtopics to make them clear. Usually, the topics are in line with the sub-topics in the methodology. The flow/story is missing in the results. This comment applies to the whole results section, Please revise.

(2)   Table 5- third column- interpret rav

(3)   Figure 5 and Figure 6- Please combine them and show it as a comparison of CTAC and Se-CTAC molecular complex. Please illustrate the figure well ordered.

(4)   Figure 8- The graphs must be labelled separately and it should be mentioned in the figure description.

(5)   Line 317- unusual writing pattern. Revise the sentence.

(6)   Figure 9 – This figure is not informative. Why did you label the plates 1,2,3,4 and 5? Did you prepare dilutions? Where are the triplicates of plates? Revise the figure description.

(7)   Line 329- A dilution series should be made if you can’t count the colonies. Please re-design the microbiological experiments and perform them again. Ignoring this will lead to a serious mistake.

(8)   The results interpretation of the microbiological work is not acceptable at this stage, Please refer to standard methods in the literature.

(9)   Line 329-343 – Detailed description of colony morphology is not necessary here. The authors have not taken a colony count and analysed the effect of antimicrobial properties based on that. Vague responses such as “vigorous growth”, and “moderate growth” should be avoided in the description. Colony counts (CFU/ml) should be taken in each dilution. Dilutions which give more than 300 colonies should be excluded and colonies 30-300 plates should be examined to determine the antimicrobial properties of a compound.

(10)      Also, I strongly suggest performing MIC (Minimum inhibitor concentration) test for the compounds such as Se NPs-CTAT and etc.. Please refer to a method and include the results in this manuscript. Without performing standard tests like MIC or disk diffusion test, you cannot argue/comment about the antimicrobial properties of your compound.

(11)      Line 346 – What do you mean by sample numbers 1,2,3,4,5? Please elaborate on where it appears in the first place

(12)      Line 359- weak culture- Vague response- Not acceptable

(13)      Figure 10 and Lines 391-393 – How can you confirm the mode of action of the compounds from the basic and inaccurate microbiological work you have carried out? How do you confirm that your compounds attacked/damaged the cell membrane or cell wall? You have not even remotely carried out this in your study. It can be only considered as an assumption or a hypothesis. The way authors have written this information is misleading. Therefore, you cannot mention “Considering the results of other investigators and the results of this study, we visualized the mechanism of antibacterial action…” Please remove what you have not performed.

(14)      Line 398 – What do you mean by the “The resulting scheme”? Please revise the sentence in a meaningful way.

(15)      Line 398-402 - Once again, how do you confirm/say your Se NPs-CTAC molecular complex degrades the proteins and polysaccharides in the microbial cell structure? You have no evidence to conclude this. The authors have not done the experiments to facilitate this statement, Please revise.

Discussion

(1)   I assume the authors have discussed the findings in the results section. However, the authors need to discuss the results separately under the discussion. Please see the journal template for nanomaterials.

(2)   I suggest the authors include recent references to support your findings.

(3)   I invite ALL the authors to make a significant contribution to the discussion part.

Conclusion

(1)   Line 417, 426- Please avoid using terms like “we”

(2)   Line 430-432 – Once again, you have insufficient data to conclude antibacterial properties. Please refer to reliable, precise microbiological methods in the literature and perform the experiments again.

(3)   Line 432-434 – Statements like this should be written with the terms “could be” or “might be”, Please revise.

(4)   Conclusion is quite long. Please revise.

(5)   Please add more future aspects/ work.

Author Response

Dear Reviewer 3,

We are grateful for your positive evaluation and for the time devoted to review our manuscript. All comments were useful and pleased us with the high level of understanding of the topic. We have addressed all recommendations as requested. All changes in the manuscript are marked by green. Please see the point-by-point answers below

Title: Please correct the spelling “cethyltrimethylammonium”

Thank you for attentiveness. This was fixed and corrected

Abstract

(1)   Introductory sentences lack in the abstract portion. Please include them. Thank you for suggestion. Introductory sentences were added to Abstract.

(2)   Line 27,37- Grammar errors, please revise.

Abstract was full revised

(3)   Line 39 – Shorten the sentence.

Abstract was full revised

(4)   Line 40 – Specify the broad applicability.

Abstract was full revised

(5)   The abstract portion exceeds the word count (approximately 200), please revise.

Abstract was full revised

Introduction

(1)   Line 47- What are the severe issues of “Se insufficiency”? Address them. added

(2)   Line 50- What is meant by “other forms”, revise the sentence.

Thank you for comment. The sentence was revised

(3)   After line 50, the authors jumped into the synthesis part, Please address the research question in detail before that.

Thank you for suggestion. New sentences were added

(4)   Line 67- Abbreviate QAS.

Corrected

(5)   Line 69- Incomplete sentence, revise.

Corrected

(6)   Overall, the introduction part is not sufficient, it needs to be more descriptive and supported with recent references. And it should present the novelty of the research. Please revise.

Corrected

Method

(1)   Lines 87,90,133, 157 167- Avoid using terms such as “First, Next, Then” in scientific writing. This applies to the whole manuscript.

Thank you for recommendation. This was corrected

(2)   Why did you use simple present tense? *Methodology section should be written in past passive tense. Line 87, 92,93,120 Revise

Revised

(3)   Line 98, 223, 250- Revise all the figure titles.

Ex: Please write Figure 1. instead of “Fig.1”

Revised

(4)   Line 107- Confirm if the name of the software appears correctly in the text.

It is correct

(5)   The methodology should contain a separate section to present/say about the precise statistical analysis performed in the study if applicable.

Thank you for suggestion. Corresponding subsection was added

(6)   Table 1- What do you mean by “Variation levels of parameters”? Does that mean different concentrations? Please revise titles.

Revised

(7)   Table 3 – Why don’t the pH values have standard deviations? Not taking replicate values is a serious mistake. Please re-do and revise

In this table and corresponding paragraph, the method of preparation of Robinson-Britton buffer solutions according to the standard method is presented. According to the method, buffer solutions are obtained with a well-defined pH. After preparation of buffer solutions, pH was additionally measured using a pH meter. Deviations related to the error of the analytical equipment used have been added to the main text.

(8)   Line 118-Avoid repetition, revise

Corrected

(9)   It seems like the authors have not characterized the Se NPs stabilized by CTAC using proper methods. It is very important in introducing a novel nano compound. Please perform the following and any other techniques in the literature:

UV-visible spectroscopy

Fourier-transform infrared spectroscopy (FTIR)

EDX spectrum

Thank you for recommendation. We added spectra of FTIR and UV-visible spectroscopy.  

Zeta potential analysis

Zeta potential is shown in Figure 4

(10)      Line 133-139 Correct the chemical formulas (subscript). This applies to the whole document

Corrected

(11)      Line 137- What is x ml here? Give a value or interpret here

Depending on the added amount of sodium hydroxide, the resulting solution acquires the required pH value

(12)      Line 143 – Avoid using terms such as “we” and “I” in scientific writing Corrected

(13)      Line 149, 177- Revise the sentence. You do not have to write the word bacteria or mold at the end of the strain name.

Corrected

(14)      How did you sterilize the medium? Filter sterilization or autoclave? Sentence is confusing, Please revise

Revised

(15)      Line 166- Culture media composition better to be included in a supplementary document.

Thank you for recommendation. The main stages of preparation of nutrient media and cultivation of Escherichia coli, Micrococcus luteus and Mucor were moved to Supplementary.

(16)      Line 172-175– These are very basic microbiological steps. Too much information, Please simplify the sentences, Revise

Revised. Information was moved to Supplementary.

(17)      Line 184-193- Very basic microbiological steps, No need to elaborate this much, it is unnecessary, Please shorten the sentences.

Shortened. Information was moved to Supplementary.

(18)      Line 195-196 Give the incubation times for bacteria and mold separately. In the preparation procedure, autoclaving parameters are indicated separately for each medium. Incubation conditions were added to the main text.

(19)      Section 2.5 -Antibacterial and fungicidal activity of Se NPs needs to be thoroughly revised. The flow is confusing. Please divide it into segments and make it neat and more understandable.

Subsection 2.5 was modified.

Ex: media preparation, cultivation of bacteria and fungi (mold) and etc. Try to avoid elaborating on very basic microbiological steps. It interferes with the main focus of that section.

Information was moved to Supplementary.

(20)      The main focus of how you tested the compound (Se NP) for antimicrobial activity is not clear at all. Failing to illustrate this is a major drawback. Please specify the method. (whether it is disk diffusion, pour-plate, spread-plate or streak?).

Thank you for recommendation. Data on the antibacterial and fungicidal activity of selenium nanoparticles studied by the disco-diffusion method have been added.

(21)      Table 4 – The data is not clearly presented. What is 1,2,3,4,5?

The numbers 1,2,3,4,5 in the table correspond to the numbers of the petri dishes, which are presented in Figure 9

Results

(1)   Results are very poorly organized and illustrated. It appears that all the results are just listed without proper order. At least, please give subtitles/subtopics to make them clear. Usually, the topics are in line with the sub-topics in the methodology. The flow/story is missing in the results. This comment applies to the whole results section, Please revise.

Thank you for the comment. This was revised

(2)   Table 5- third column- interpret rav

rav means average hydrodynamic radius of selenium nanoparticles. Definition was added under the table.

(3)   Figure 5 and Figure 6- Please combine them and show it as a comparison of CTAC and Se-CTAC molecular complex. Please illustrate the figure well ordered.

Thank you for recommendation. Figures were combined as suggested.

(4)   Figure 8- The graphs must be labelled separately and it should be mentioned in the figure description.

Corrected

(5)   Line 317- unusual writing pattern. Revise the sentence.

Revised

(6)   Figure 9 – This figure is not informative. Why did you label the plates 1,2,3,4 and 5? Did you prepare dilutions? Where are the triplicates of plates? Revise the figure description.

In the methods and materials section in Table 3, the concentrations of the precursor, reducing agent, stabilizer and Se NPs - CTAC are indicated – they are designated by numbers 1,2,3,4,5. The Petri dishes in the figure are arranged in accordance with these numbers (samples order) and concentrations.

(7)   Line 329- A dilution series should be made if you can’t count the colonies. Please re-design the microbiological experiments and perform them again. Ignoring this will lead to a serious mistake.

Antimicrobial studies were modified and supplemented by additional results.

(8)   The results interpretation of the microbiological work is not acceptable at this stage, Please refer to standard methods in the literature.

Antimicrobial studies were modified and supplemented by additional results.

(9)   Line 329-343 – Detailed description of colony morphology is not necessary here. The authors have not taken a colony count and analysed the effect of antimicrobial properties based on that. Vague responses such as “vigorous growth”, and “moderate growth” should be avoided in the description. Colony counts (CFU/ml) should be taken in each dilution. Dilutions which give more than 300 colonies should be excluded and colonies 30-300 plates should be examined to determine the antimicrobial properties of a compound.

Antimicrobial studies were modified and supplemented by additional results.

(10)      Also, I strongly suggest performing MIC (Minimum inhibitor concentration) test for the compounds such as Se NPs-CTAT and etc.. Please refer to a method and include the results in this manuscript. Without performing standard tests like MIC or disk diffusion test, you cannot argue/comment about the antimicrobial properties of your compound.

Thank you for suggestion. The study was expanded by disco-diffusion method.

(11)      Line 346 – What do you mean by sample numbers 1,2,3,4,5? Please elaborate on where it appears in the first place

These figures correspond to certain concentrations and are presented in Table 3

(12)      Line 359- weak culture- Vague response- Not acceptable

Corrected

(13)      Figure 10 and Lines 391-393 – How can you confirm the mode of action of the compounds from the basic and inaccurate microbiological work you have carried out? How do you confirm that your compounds attacked/damaged the cell membrane or cell wall? You have not even remotely carried out this in your study. It can be only considered as an assumption or a hypothesis. The way authors have written this information is misleading. Therefore, you cannot mention “Considering the results of other investigators and the results of this study, we visualized the mechanism of antibacterial action…” Please remove what you have not performed.

Thank you for your comment. This figure is a schematic hypothesis based on the results of this study and other researchers. We mentioned, that this is hypothesis and our further aim is confirming of this hypothesis.

(14)      Line 398 – What do you mean by the “The resulting scheme”? Please revise the sentence in a meaningful way.

Corrected

(15)      Line 398-402 - Once again, how do you confirm/say your Se NPs-CTAC molecular complex degrades the proteins and polysaccharides in the microbial cell structure? You have no evidence to conclude this. The authors have not done the experiments to facilitate this statement, Please revise.

Thank you for the comment. This proposal is based on previous researches of other authors. References are given below.

Discussion

(1)   I assume the authors have discussed the findings in the results section. However, the authors need to discuss the results separately under the discussion. Please see the journal template for nanomaterials.

Thank you for suggestion. We usually combine Results and Discussion in one section, e.g. in our previous work published in Nanomaterials (https://doi.org/10.3390/nano13091577). According to Instruction for authors https://www.mdpi.com/journal/nanomaterials/instructions Discussion can be combined with Results. Thus, we would like to leave the structure of the manuscript as it is and believe that this will not influence on the quality of the work.   

(2)   I suggest the authors include recent references to support your findings.

Thank you for suggestion. We added several new recent references.  

(3)   I invite ALL the authors to make a significant contribution to the discussion part.

Thank you for suggestion. Results and Discussion section was modified with contribution of all authors.

Conclusion

(1)   Line 417, 426- Please avoid using terms like “we”

Corrected

(2)   Line 430-432 – Once again, you have insufficient data to conclude antibacterial properties. Please refer to reliable, precise microbiological methods in the literature and perform the experiments again.

Results of CFU counting and disco-diffusion method were added in Conclusions

(3)   Line 432-434 – Statements like this should be written with the terms “could be” or “might be”, Please revise.

Revised

(4)   Conclusion is quite long. Please revise.

Revised

(5)   Please add more future aspects/ work.

Added

Round 2

Reviewer 1 Report

Comments and Suggestions for Authors

All the comments have been addressed in the revised manuscript.

Author Response

Dear Reviewer 1, we appreciate your work and positive evaluation of the revised manuscript. 

Reviewer 3 Report

Comments and Suggestions for Authors

I thank the authors for taking careful consideration during the revisions. The authors should include the page and line numbers of the revised text in the next revision round for the reviewers to track changes. The revisions that you make should be visible in both the review report and in the revised manuscript. Please pay attention to all the comments and suggestions very carefully from the reviewer and address each one by one.

Abstract (Page 1)

(1) Line 28- What is sol? Give an explanation or correct

(2) Line 31-32- Grammar errors, please revise

(3) Line 33-34- This is an inaccurate statement; Did you do clinical trials? You can only say, Se NPs – CTAC has antimicrobial properties, revise.

(4) Line 35- You have to specify this conclusive line

Introduction (Page 2)

(1) Line 5- Elaborate your research question more, still a bit unclear. Revise. Try to cite literature

(2) Line 10- what is bold? Revise

(3) Line 13- bacteriophages are not bacteria! False statement, revise

(4) Line 27- Grammer errors, please revise

(5) Overall, the introduction part is not sufficient. It is cited. But previous/ recent work carried out by other researchers should be properly cited, Please do that.

Method (Page 2)

(1) Line 53- Please use this as the title for Figure 1

(Page 3)

(2) Lime 18- grammar error, please revise

(Page 4)

(3) Line 4-9- While you need to provide how you characterized the NP, please try to concise this paragraph. Or it may appear like as too much information.

(4) Line 23- Table 2- This looks like raw data, Please state the reason, what you determined through these pH values

(5) Line 38-40- It seems like the authors have ignored my comment in the first round. I am not requesting the basic microbiological steps like media preparation. You need to state the exact methodology for testing the NP for antimicrobial activity. You need to state the volumes and all that. You don’t assess antimicrobial activity by just observing some growth, it has to be precise data like colony counts in the range of 30-300 for bacteria, and 15-150 for fungi and eventually converted into CFU/ mL. And your data should be supported

with controls. If you fail to present this data, it will be considered unacceptable microbial data. Revise

(Page 5)

(1) Line 2- Once again, this shows that colony numbers are too high (>300). That’s why the authors need to select the plates between 30-300. It has not been done; I assume. If you took this colony count, it is unacceptable. The title itself mentions the antimicrobial activity. But, the microbiological work conducted here is very inadequate. The authors should get the advice of a microbiologist and re-do the section 2.5.

(2) Line 9- disco diffusion? Please revise

(3) Line 9-13 This flow of this section is very unclear. Please revise, Please refer to reliable research papers for the corrections.

(4) Line 11- Substance? name it. Certain concentration? Include in the text, Please revise.

(5) Line 12-transplanted culture? Awkward text. Prefer inoculated culture

(6) Line 13-suppression zones? Or inhibition zones? Please revise

Results & Discussion

(Page 5)

(1) Line 28- Table 4- What is meant by sample 1 to 9? Please indicate in the text. You can prepare graphs using this kind of data.

(Page 7)

(2) Line 6- Figure 4-Please enhance the scale bar of the TEM images. And label the figures as Figure 4a , 4b etc.

(Page 8)

(1) It is good that authors combined the figure 5 and 6. Easy to understand the differences.

(Page 9)

(1) Since you have decided to present the results and discussion in one segment, you have to add recent relevant findings as citations in the text. Please do that.

(Page 10)

(2) Line 2- Figure 7- The images are not labelled properly. Label each and every image in a figure with subtitles in the figure description. This applies to all the figures.

(3) Line 16- What are sols? Write the full word in the text.

(Page 11)

(1) Line 2- Figure 8- same as Figure 7, revise

(2) Line 25- Omit the word plot, just say figure 8

(Page 12)

(1) Figure 9- Why is the axis of this graph not properly labelled? They are just symbols? How can the reader understand? This apply to all the graphs?

(2) Line 9- (Δ = 6 nm) Have you abbreviated this symbol before in the text? Revise

(Page 13)

(1) Figure 10- This figure is not informative, As I have commented the last time, Please answer the following questions.

(i) Did you prepare dilutions when doing this?

(ii) Did you do triplicate plating? It looks like you have not, according to Table 5.

(iii) 1,2,3,4,5- needs to be explained again in the figure title.

(iv) You say that these Petri dishes are arranged in accordance with these numbers (sample order) and concentrations. But there is no mention in the figure title.

(2) Table 5- Once again, I have to say these results are inaccurate and cannot be acceptable. Solid growth- vague response- How could you assess the antimicrobial properties of a compound with this type of vague results? And the colony counts are not even with scientific notations.

It should be like- 6.12 x 103 CFU/ mL. Please refer to literature and re-do the micro experiments correctly. Authors keep ignoring this is a serious mistake and can have its consequences.

(Page 14)

(1) Line 10- “between 100 and 1000”. You do not take colonies >300 into consideration. This is inaccurate.

(Page 15)

(2) Line 14, 17- inhibition zones are not with standard deviation/error values. Why is that? That result can be tabulated. Revise.

(Page 16)

(1) Line 18- You mentioned the potential of this formulation for practical application without even carrying out the necessary and accurate microbiological steps. This is not acceptable. Please pay attention to the comments and previous studies and how the antimicrobial studies are done. Please revise.

Author Response

We are thankful to Reviewer 3 for professionalism and willingness to help to improve the quality of our manuscript. Sorry for inadequate decision of some comments on the previous stage. Now, after careful revision of the manuscript with additional study of recent works on the topic and carried out supplemental experiment, we are ready to submit the revised version of our work. All changes were marked by blue in the text and belong only to comments and recommendations of Reviewer 3. However, as was requested, we added references to lines and pages of modified parts of the text in the Response to Reviewer document. Please find Response to Reviewer in attach file below.      

Round 3

Reviewer 3 Report

Comments and Suggestions for Authors

Reviewer comments and suggestions

            The authors should include the page and line numbers of the revised text in the next revision round for the reviewers to track changes. The revisions that you make should be visible in both the review report and in the revised manuscript.  Please pay attention to all the comments and suggestions very carefully from the reviewer and address each one by one.

(Page 1)

Line 28 – sol? Interpret, revise. This comment was given in a previous round. Do not ignore.

Line 31- considering the model organisms that the authors chose for antimicrobial activity; why did you choose Mucor? Not even mycotoxin has been isolated from Mucor which is basically a harmless organism. If you are choosing an organism for an antimicrobial-related test, at least the model organism should possess some pathogenicity. Otherwise, what is the point? This should be addressed as a limitation in this study.

Line 51- Again, the authors talk about the problem of multidrug-resistant bacterial strain, yet your compound Se NPs – CTAC was not even tested for major bacterial or fungal pathogens! In that case, the aim and the experiment flow are contradictory. This is a significant limitation of this study.

(Page 4)

Line 39- The topic says fungicidal activity. Yet you have only tested the compound for a single non-pathogenic fungal species. You are unable to come to conclusions about a fungicidal activity based on that. The compound should at least be tested for 4 bacterial and 4 fungal pathogens.

(Page 5)

Line 7-10- did you do McFarland standards? Why is this information included now, but not in the first version of the manuscript? The preparation of  McFarland standard should be included in the supplementary doc.

Line 9,10- 150 × 103 CFU/ml should be in scientific notation. This applies to the whole document. Revise.

Line 15- Mention the dilution series that you prepared, which dilutions? Revise

Line 16- If you perform triplicates, include the images in the supplementary document with the serial dilutions. And label them accurately.

Why did you move Figure (S1) to the supplementary document? It used to be in the main text. Figure S1 does not depict any dilutions or triplicates. It is a raw image without proper details. The authors were given comments and suggestions from the previous round about this experiment to re-do it. The authors have ignored this comment for the second time and have not performed the accurate microbiology test. Without this accurate microbiology test, this paper cannot be published.

(Page 13)

Figure 10- Authors mention about the disk diffusion test. But where are the 2 mm disks on the plate? And inhibition zones are not visible in the plates although Table 6 reports inhibition zones. Did you perform the disk diffusion test in replicates? Please include some images of the replicates in the supplementary documents. Revise.

(Page 16)

Line 26- How do you say/confirm that Se NPs - CTAC have significant antibacterial properties? Did you statistically prove that? It is essential to give statistical significance in research! Without a statistical approach, you can’t say it is significant by just observing raw data! Please analyse them using a statistical software- This suggestion is highly recommended. (Tip-Find significant difference using a p-value (95% level), revise

Line 31-32- Spell check, revise.

Line 25- “sol”?

*Overall, the manuscript contains raw data. They can be statistically analysed. The data representation is poor.

Suggestion: arranging graphs and other data representation methods, rather than indicating raw values and numbers in tables. So the reader can grasp the results meaningfully and quickly.

Author Response

Dear reviewer, 

The revision of the manuscript was carried out. Please find point-by-point responses in the attached document.  

Round 4

Reviewer 3 Report

Comments and Suggestions for Authors

Authors have improved manuscript.

Table 1-7

please use nanomaterials template. The  title of 'Table x' starts at 4.6 cm from left side of margin.  The  'Table x' should be bold letter. 

There are three solid lines top line, second line and bottom line without vertical line and other middle line.

Table 1. This is a table. Tables should be placed in the main text near to the first time they are cited.

Title 1

Title 2

Title 3

entry 1

data

data

entry 2

data

data 1

Figure 1,2,3,4,6,8,9,10,11,12 

The  figure legend 'Figure x' starts at 4.6 cm from left side of margin.  The  'Figure x' should be bold letter. 

Figure 5 and 7 are large figure.

The figure legend 'Figure 5' starts at 0 cm from left side of margin. The  'Figure 5' should be bold letter. 

Subtitle should be started at 4.6 cm from left side of margin

2.1 Synthesis of Se NPs - CTAC 

2.2 Optimization of the synthesis of Se NPs - CTAC

2.3 Characterization of Se NPs - CTAC

2.4 Stability of Se NPs - CTAC

2.5 Potential antibacterial and fungicidal activity of Se NPs - CTAC

2.6 Statistical data processing

3.1 Optimization of parameters for the synthesis of Se NPs - CTAC

3.2 Transmission electron microscopy of Se NPs – CTAC

3.3 Computer quantum chemical modeling of CTAC molecule and Se NPs-CTAC molecular complex

3.4 Spectral characteristics of Se NPs - CTAC

3.5 Stability of Se NPs - CTAC at different pH

3.6 Stability of positive and negative Se NPs sols at various ions 

Author Response

We appreciate the Reviewer 3 for a great work have been done that helped to improve our manuscript. All comments were considered and decided consequently. 

Table 1-7

please use nanomaterials template. The  title of 'Table x' starts at 4.6 cm from left side of margin.  The  'Table x' should be bold letter. 

Thank you, corrected

There are three solid lines top line, second line and bottom line without vertical line and other middle line.

Table 1. This is a table. Tables should be placed in the main text near to the first time they are cited.

Title 1

Title 2

Title 3

entry 1

data

data

entry 2

data

data 1

Thank you, corrected

Figure 1,2,3,4,6,8,9,10,11,12 

The  figure legend 'Figure x' starts at 4.6 cm from left side of margin.  The  'Figure x' should be bold letter. 

Thank you, corrected

Figure 5 and 7 are large figure.

The figure legend 'Figure 5' starts at 0 cm from left side of margin. The  'Figure 5' should be bold letter. 

Thank you, corrected for both figures 5 and 7

Subtitle should be started at 4.6 cm from left side of margin

2.1 Synthesis of Se NPs - CTAC 

Thank you, corrected

2.2 Optimization of the synthesis of Se NPs - CTAC

Thank you, corrected

2.3 Characterization of Se NPs - CTAC

Thank you, corrected

2.4 Stability of Se NPs - CTAC

Thank you, corrected

2.5 Potential antibacterial and fungicidal activity of Se NPs - CTAC

Thank you, corrected

2.6 Statistical data processing

Thank you, corrected

3.1 Optimization of parameters for the synthesis of Se NPs - CTAC

Thank you, corrected

3.2 Transmission electron microscopy of Se NPs – CTAC

Thank you, corrected

3.3 Computer quantum chemical modeling of CTAC molecule and Se NPs-CTAC molecular complex

Thank you, corrected

3.4 Spectral characteristics of Se NPs - CTAC

Thank you, corrected

3.5 Stability of Se NPs - CTAC at different pH

Thank you, corrected

3.6 Stability of positive and negative Se NPs sols at various ions 

Thank you, corrected